# Use of parenteral nutrition in the first postnatal week in England and Wales: an observational study using real-world data

James Webbe [1], Cheryl Battersby [1], Nicholas Longford,[1] Kayleigh Oughham,[2] Sabita Uthaya,[1] Neena Modi,[1] Chris Gale [1]

¹Neonatal Medicine, Imperial College London, London, UK
²Neonatal Data Analysis Unit, Imperial College London, Faculty of Medicine, London, UK

**Correspondence to**
Dr Chris Gale; christopher.gale@imperial.ac.uk

## ABSTRACT

**Background** Parenteral nutrition (PN) is used to provide supplemental support to neonates while enteral feeding is being established. PN is a high-cost intervention with beneficial and harmful effects. Internationally, there is substantial variation in how PN is used, and there are limited contemporary data describing use across Great Britain.

**Objective** To describe PN use in the first postnatal week in infants born and admitted to neonatal care in England, Scotland and Wales.

**Method** Data describing neonates admitted to National Health Service neonatal units between 1 January 2012 and 31 December 2017, extracted from routinely recorded data held the National Neonatal Research Database (NNRD); the denominator was live births, from Office for National Statistics.

**Results** Over the study period 62 145 neonates were given PN in the first postnatal week (1.4% of all live births); use was higher in more preterm neonates (76% of livebirths at <28 weeks, 0.2% of term livebirths) and in neonates with lower birth weight. 15% (9181/62145) of neonates given PN in the first postnatal week were born at term. There was geographic variation in PN administration: the proportion of live births given PN within neonatal regional networks ranged from 1.0% (95% CIs 1.0 to 1.0) to 2.8% (95% CI 2.7 to 2.9).

**Conclusions and relevance** Significant variation exists in neonatal PN use; it is unlikely this reflects optimal use of an expensive intervention. Research is needed to identify which babies will benefit most and which are at risk of harm from early PN.

**Trial registration number** ClinicalTrials.gov: NCT03767634; registration date: 6 December 2018.

### WHAT IS ALREADY KNOWN ON THIS TOPIC
⇒ Parenteral nutrition is commonly given to neonates unable to receive adequate milk feeds or while milk feeds are introduced.
⇒ There is limited evidence to inform parenteral nutrition use in neonates.

### WHAT THIS STUDY ADDS
⇒ The early use of parenteral nutrition is common; 17% of all babies admitted to neonatal units receive parenteral nutrition (PN) in the first postnatal week.
⇒ Fifteen per cent of neonates given PN in the first postnatal week are born full term.
⇒ There is variation in the use of parenteral nutrition in the first week of postnatal life between regional neonatal networks.

### HOW THIS STUDY MIGHT AFFECT RESEARCH, PRACTICE OR POLICY
⇒ This study provides data to help in the planning of future PN research in neonates.
⇒ The unexplained geographic variation in PN use we have identified suggests that resources are being wasted and neonates are recieving suboptimal care.

## INTRODUCTION

In 1968, parenteral nutrition (PN) was used to support the metabolic needs of a term neonate with small bowel atresia.[1] Following this, PN has been increasingly used to supplement the nutrition of sick or preterm neonates. The widespread use of PN has been encouraged on the basis that optimising nutrition will improve short and long-term outcomes.[2] It is considered most beneficial for neonates born preterm or with lower birth weight who have fewer reserves and may accrue large nutritional deficits before enteral feeds are established.[2] Despite widespread use, the impact of PN on key neonatal outcomes has not been evaluated in randomised controlled neonatal trials powered for clinically meaningful and functional end-points.

Therefore, while effects on short-term biochemical markers such as nitrogen balance are well described,[3] evidence to support beneficial effects on survival and neurodevelopment are lacking.[3 4] Conversely, PN carries well-described risks, of which the most serious and common is bloodstream infection.[5] Recent evidence from large randomised controlled trials in critically unwell adults[6] and children[7] showed that use of PN during

**Table 1** Gestational age of neonates receiving PN in the first postnatal week as a proportion of total live births

| Gestational age category* at birth | Neonates receiving PN in the first postnatal week by year of birth n (% of live births) | | | | | |
|---|---|---|---|---|---|---|
| | 2012 | 2013 | 2014 | 2015 | 2016 | 2017 |
| Extremely preterm | 2317 (72) | 2309 (75) | 2267 (77) | 2348 (78) | 2421 (76) | 2325 (75) |
| Very preterm | 3493 (61) | 3900 (71) | 3896 (71) | 4059 (73) | 4135 (73) | 4110 (74) |
| Moderate and late preterm | 2343 (5.3) | 2640 (6.1) | 2796 (6.4) | 2683 (6.0) | 2547 (5.6) | 2375 (5.2) |
| Term | 1370 (0.2) | 1688 (0.3) | 1618 (0.3) | 1528 (0.2) | 1484 (0.2) | 1493 (0.2) |
| Total | 9523 (1.3) | 10 537 (1.5) | 10 577 (1.5) | 10 618 (1.5) | 10 587 (1.5) | 10 303 (1.5) |

Extremely preterm: <28+0 weeks, very preterm: 28+0–31+6 weeks, moderate and late preterm; 32+0–36+6 weeks, term >36+6 weeks.
Number in brackets indicates the percentage of all live births given PN in each category (Denominator data from ONS birth characteristics in England and Wales).
Neonates with missing data for gestational age=3.
*Gestational age at birth categorised using WHO definitions.[41]
ONS, Office for National Statistics; PN, parenteral nutrition.

the first 7 days of admission to an intensive care unit led to worse outcomes, when compared with delayed PN administration, indicating that the harms of early PN outweigh benefits in these populations. Although there has not been a similar trial in neonatal care, subgroup analysis in the PePaNIC trial of term neonates looked after on paediatric intensive care units also showed increased rates of nosocomial infection with early PN use,[8] suggesting that early PN use should be targeted at neonates with most potential for benefit.

Given the uncertain balance of risk and benefit for neonatal PN use, it is unsurprising that international practice is variable: some neonatal units in high-income countries provide PN to up to 70% of neonatal admissions,[9] while others report not using PN.[10] In the Great Britain, a 2011 report from the National Confidential Enquiry into Patient Outcome and Death found considerable variation in neonatal management in 2008 with only 24% of

patients receiving PN that was considered best practice.[11] Following this, a national framework[12] and National Institute for Health and Care Excellence (NICE) guidance[13] have been developed. The most recent NICE guidance makes recommendations about prescription, administration, monitoring and recipients of PN in neonatal units; recommending all neonates born before 30+0 weeks+days gestation or weighing under 1250 g at birth, and any who are unable (or not expected) to establish milk feeds of ≥100 mL/kg/day by postnatal day 5, receive early PN.

We aimed to describe how PN is used in the first postnatal week, to explore how use is influenced by gestational age, birth weight, geographical region and to compare how use has changed over time in the period prior to the publication of 2020 NICE guidance.

## OBJECTIVE

To describe the pattern of PN use in neonatal units in England, Scotland and Wales in the first 7 postnatal days.

## METHODS
### Study design

This study was an epidemiological description of practice: we preregistered it (Clinicaltrials.gov) and published the study protocol.[14] We report it in line with REporting of studies Conducted using Observational Routinely-collected Data (RECORD) guidelines.[15]

### Data source

We used deidentified data held in the National Neonatal Research Database (NNRD).[16] The NNRD holds data extracted from electronic health records completed by health professionals during routine clinical care.[17] The Neonatal Data Set, a defined national data standard[18] comprising approximately 450 items, is extracted and transmitted to the Neonatal Data Analysis Unit at Imperial College London. The NNRD holds data from all neonates admitted to National Health Service (NHS) neonatal

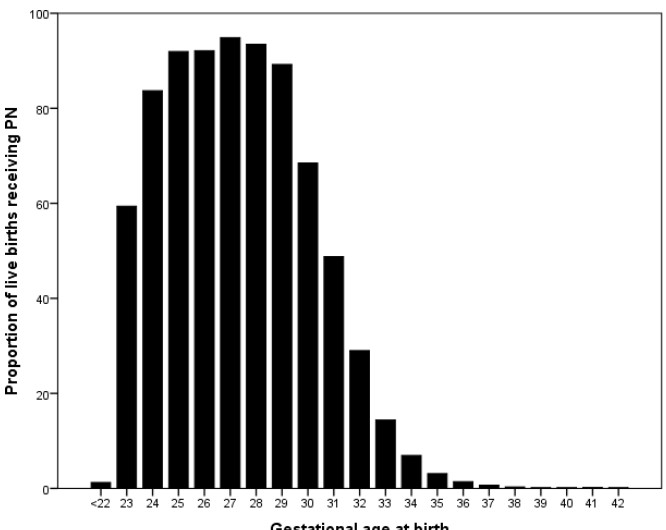

**Figure 1** Proportion of live births receiving PN during the first postnatal week of life from 2012 to 2017. All births included from 2012-2017. Total number of births = 4,196,314 Neonates with missing data for gestational age = 3.

**Table 2** Birth weight of neonates receiving PN in the first postnatal week as proportion of neonatal unit admissions

| Birth weight at birth* | Neonates receiving PN in the first postnatal week by year of birth n (% of neonatal units admissions) | | | | | |
|---|---|---|---|---|---|---|
| | 2012 | 2013 | 2014 | 2015 | 2016 | 2017 |
| <1 kg | 2401 (*92*) | 2484 (*94*) | 2442 (*95*) | 2542 (*95*) | 2547 (*96*) | 2541 (*96*) |
| <1.5 kg | 5591 (*77*) | 5878 (*82*) | 5936 (*83*) | 6090 (*84*) | 6213 (*85*) | 6095 (*85*) |
| <2.5 kg | 7999 (*27*) | 8679 (*29*) | 8756 (*29*) | 8880 (*28*) | 8935 (*28*) | 8628 (*28*) |
| >2.5 kg | 1524 (*3.1*) | 1857 (*3.5*) | 1821 (*3.2*) | 1738 (*2.8*) | 1652 (*2.6*) | 1675 (*2.4*) |

Number in brackets indicates the percentage of all neonates admitted to a neonatal unit given PN in each category (Denominator data from NNRD).
Neonates with missing data for birth weight=4.
*Birth weight categorised using WHO definitions.[42]
PN, parenteral nutrition.

units in England, Scotland and Wales; in total, the NNRD contains data from about one million neonates from 2008 to the present. Accuracy and completeness of NNRD data have been confirmed by comparison with Case Record Forms from a prospective clinical trial, which showed high data completeness and accuracy (>95%).[19] Data for this study were extracted by author KO, operating within the guidelines established by these approvals; no other investigators had accessed the wider NNRD for this study. No data cleaning methods were required for this study, and no data linkage was required. We obtained population-level data for total live births by gestational age and birth weight from the Office for National Statistics (ONS)[20–25] and for live births by neonatal network from Mothers and Babies: Reducing Risk through Audits and Confidential Enquiries across the UK (MBRRACE-UK) reports,[26–29] for denominator data. Population-level data for total neonatal unit admissions by gestational age and birth weight were obtained from the NNRD for denominator data.

### Participants

The study population was all neonates born between 1 January 2012 and 31[27] December 2017 and admitted to a neonatal unit in England and Wales. There were no exclusion criteria: this was intended to maximise the sample size and ensure complete, population-level data.

### Variables

The primary outcome was any use of PN in the first 7 postnatal days. To describe the background characteristics of neonates given PN, we extracted data relating to gestational age at birth, birth weight, year of birth and neonatal network[30] of birth. The variables extracted from the NNRD, and how they were defined, are listed in online supplemental eTable 1.

### Statistical methods

We described the characteristics of neonates that received PN in the first 7 postnatal days and compared use between different groups. For gestational age and birth weight, we grouped neonates according to well-established and widely accepted WHO categories.[31 32] To explore differences in PN use across geographical regions, we grouped neonates by neonatal network of birth.[30] We compared changes in PN prescribing over time by grouping neonates according to the year of birth.

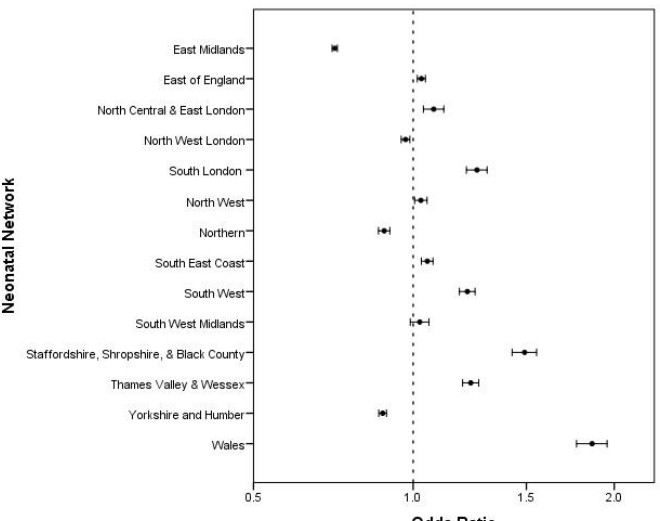

**Figure 2** Forest plot of live births receiving PN during the first postnatal week of life in each neonatal network. Point indicates odds ratio of receiving PN when compared to the national average (National average= 1.4% of live births received PN in first postnatal week). Error bars indicate 95% confidence interval for odds ratio.

### RESULTS

The only deviation from the protocol is related to the use of data from neonatal units in Scotland. We were unable to use data on babies from Scottish neonatal units due to difficulties obtaining the institutional approvals required for Public Benefit and Privacy Panel for Health and Social Care[31] approval. The project was completed using data from neonatal units in England and Wales only.

Over the 6-year study period, 4 196 314 neonates were born in England and Wales. Of these, 347 959 neonates were admitted to NHS neonatal units and had data

**Table 3** Neonates receiving PN in the first postnatal week by Neonatal Network as proportion of live births

| Neonatal network at birth | Babies receiving PN in the first postnatal week by year of birth n (% of live births) | | | | | |
| --- | --- | --- | --- | --- | --- | --- |
| | 2012 | 2013 | 2014 | 2015 | 2016 | 2017 |
| East Midlands* | 608 (†) | 601 (*1.1%*) | 607 (*1.1%*) | 553 (*1.0%*) | 570 (*1.0%*) | 566 (*1.0%*) |
| East of England | 953 (†) | 973 (*1.4%*) | 1052 (*1.5%*) | 942 (*1.4%*) | 961 (*1.4%*) | 926 (*1.4%*) |
| London—North Central & East | 771 (†) | 733 (*1.4%*) | 806 (*1.5%*) | 842 (*1.6%*) | 845 (*1.6%*) | 845 (*1.6%*) |
| London—North West | 417 (†) | 429 (*1.4%*) | 421 (*1.3%*) | 403 (*1.3%*) | 443 (*1.4%*) | 409 (*1.3%*) |
| London—South | 725 (†) | 723 (*1.6%*) | 739 (*1.7%*) | 804 (*1.8%*) | 828 (*1.8%*) | 764 (*1.7%*) |
| North West | 1233 (†) | 1206 (*1.5%*) | 1222 (*1.5%*) | 1231 (*1.5%*) | 1130 (*1.3%*) | 1031 (*1.2%*) |
| Northern | 326 (†) | 398 (*1.2%*) | 397 (*1.2%*) | 408 (*1.3%*) | 413 (*1.3%*) | 363 (*1.1%*) |
| South East Coast | 681 (†) | 723 (*1.5%*) | 707 (*1.5%*) | 720 (*1.5%*) | 717 (*1.5%*) | 710 (*1.5%*) |
| South West | 849 (†) | 805 (*1.7%*) | 785 (*1.6%*) | 839 (*1.8%*) | 813 (*1.7%*) | 786 (*1.6%*) |
| Southern West Midlands | 425 (†) | 426 (*1.3%*) | 443 (*1.4%*) | 431 (*1.4%*) | 484 (*1.5%*) | 454 (*1.5%*) |
| Staffordshire, Shropshire and Black Country | 495 (†) | 534 (*2.1%*) | 527 (*2.1%*) | 553 (*2.3%*) | 479 (*1.9%*) | 499 (*2.0%*) |
| Thames Valley and Wessex | 1062 (†) | 1068 (*1.9%*) | 1021 (*1.7%*) | 1052 (*1.8%*) | 1045 (*1.7%*) | 1034 (*1.8%*) |
| Wales | 166 (†) | 933 (*2.9%*) | 910 (*2.9%*) | 881 (*2.8%*) | 846 (*2.7%*) | 888 (*2.8%*) |
| Yorkshire and Humber | 692(†) | 820 (*1.2%*) | 791 (*1.2%*) | 795 (*1.2%*) | 847 (*1.3%*) | 878 (*1.3%*) |

Number in brackets indicates the percentage of all live births given PN in each Neonatal Network (Data from MBRRACE-UK reports).
Neonates with missing data for neonatal network=202.
*East Midlands Operational Delivery Network previously the separate Central and Trent Networks.
†MBRRACE-UK report not produced for 2012.

held within the NNRD; 62 145 were recorded as having received PN during the first postnatal week. This equates to just over 1% of all live births (table 1) and 17% of all neonatal unit admissions (online supplemental eTable 2). PN was given to neonates of all gestational ages, with neonates born preterm more likely to be recipients. However, a large proportion of the neonates who receive PN were more mature: 15% of neonates who received PN in the first postnatal week over the study period were born at term.

The proportion of live births who receive PN in the first postnatal week is lower in neonates born at 22 or 23 weeks, rises to over 90% for neonates born from 25 to 28 weeks of gestation before falling again (figure 1), (online supplemental eTable 3).

Neonates born with lower birth weights were more likely to receive PN in the first postnatal week (table 2, online supplemental eTable 4).

The use of PN differed across networks, with a range from 1.0% to 2.8% of all live births given PN in the first postnatal week (figure 2) (table 3). Rates of PN administration within neonatal networks varied less over time than the differences seen between networks.

## DISCUSSION

In this work, we described the characteristics of neonates born in England and Wales who receive PN in the first postnatal week. PN is a common intervention on neonatal units and is given to 17% of all admissions. While PN use is higher in neonates born prematurely and with lower birth weight, a considerable proportion is born at term. We also show that use varies between different neonatal networks.

That higher rates of PN use are seen in neonates born more preterm is unsurprising, as these populations are considered most likely to benefit. The lower rates seen in the most preterm neonates (those born at <26 gestational weeks) are likely because many of these babies die before admission to a neonatal unit and before PN is commenced. However, because of the much larger proportion of babies born at more mature gestations, it is noteworthy that around 15% of all babies given PN are born at term (although this still means that only 0.2% of term births receive PN). The energy requirements,[32] indications for use and metabolic stability of these groups differ, thus the risks and benefits of PN may also differ substantially across gestational ages: guidelines and practice appropriate in one group may not be optimal in differing populations.

We found that PN use in the first postnatal week varied significantly between neonatal networks. This is in keeping with variations between neonatal units in PN use in other settings.[9 10] However, we were unable to find another comprehensive population-based study of national PN use. From our descriptive analysis, it is not possible to determine whether the variation we identified is explained by regional differences in rates of prematurity, neonatal sickness or other case-mix factors

(such as congenital anomalies or neonatal surgical conditions), nor, given the paucity of evidence, is it possible to comment on what rate of PN use should be expected. However, by identifying wide variation in practice within the Great Britain, we highlight early PN use as an area where optimal practice is uncertain.[33] This variation also has financial implications. Treating a neonate with PN for 1 day in 2012 cost £37.43,[34] and if all hospitals in England and Wales treated the same proportion of live births as the highest use network, this would cost the NHS an additional £2.5 million each year. This expense may be justified if PN leads to lower mortality or morbidity, but such evidence is lacking.

The strengths of our study include the population-level coverage involving a cohort of over 4 million neonates. The population-level data meant that recruitment bias was reduced, but not fully eliminated. The NNRD covers all neonatal units in England and Wales but does not include Paediatric Intensive Care Units and surgical units that admit neonates. Some term cardiac and surgical babies will be cared for on these units and, thus, we may have underestimated the amount of PN used in these groups. We followed a prespecified protocol and data analysis plan and limited the risk of false discovery associated with multiple comparisons[35] by using the Holm-Bonferroni method. In keeping with previous studies that have used the NNRD, we had very little missing data.[19 36] Limitations of this study include that we were not able to obtain permission to use data for neonates born in Scotland, reducing the study population. As the NNRD only holds data for neonates admitted to an NHS neonatal unit for denominator data, we used data from the ONS and MBRRACE-UK to provide total numbers of live births. As MBRRACE-UK did not produce a report in 2012, this limited the number of years for which we could undertake network-level comparisons. As has been found in previous studies, due to the lack of additional information about PN in the data extracted, we are unable to describe changes in how PN was used (eg, when PN was commenced or types of PN used).[37–39]

Our findings show that PN use in the first postnatal week is common in England and Wales, with regional variation. In light of the potential harm found with early PN in critically unwell children,[7] and the lack of evidence of benefit for clinically important endpoints in infants[3 40] research is needed to ensure that PN use in neonates is underpinned by a robust evidence base. Our data will help in the planning of future trials to identify which neonates will benefit from early PN and ensure that clinical practice is based on strong evidence. Approximately, 28 neonates are started on PN each day in England and Wales: if all of these diverse patients are to receive optimal treatment, urgent research is needed to ensure that they are the neonates who will benefit most from this intervention and to avoid harm in others.

## CONCLUSION

Parenteral nutrition is commonly used in the first postnatal week across the Great Britain, with higher use in neonates born more preterm. Across all gestational age categories, no change in PN use in the first postnatal week over time was found, but there is persisting variation in use between regional neonatal networks. Research is needed to ensure that PN use in this group is well targeted.

**Acknowledgements** We wish to thank Angela Richard-Löndt and Laura Noakes, parents of preterm infants, for their support in developing this research project and BLISS for their input and support developing the dissemination plan.

**Contributors** JW, CG and NM conceived this project. JW, CG, KO and NL planned the statistical analyses. The first draft of the manuscript was written by JW and revised by NM; CG, SU, CB, NL and NM edited and reviewed the manuscript. It was approved by JW, CG, CB, KO, NL, SU and NM. CG acted as guarantor.

**Funding** The NNRD is created and maintained through grants and unrestricted funding held by NM. This includes costs involved in data transfer, storage, cleaning, merging, administration and regulatory approvals. The extraction of study data from the NNRD and analysis for this study was funded through a Mason Medical Research Fellowship awarded to JW. CG is funded by the United Kingdom Medical Research Council (MRC) through a Clinician Scientist Fellowship award.

**Competing interests** JW has received support from Chiesi Pharmaceuticals to attend an educational conference and has received a research grant from Mason Medical Research Foundation. SU has received funding from the National Institute of Health Research, the Department of Health and Prolacta Life Sciences. SU has been on the Advisory Board of Fresenius Kabi and received honoraria and travel expenses for speaking at study days organised by Fresenius Kabi. SU is a member of the National Institute for Health and Care Excellence Parenteral Nutrition Guideline Development Committee. NM is the Chief Investigator for the National Neonatal Research Database. In the last 5 years, NM has served on the Board of Trustees of the Royal College of Paediatrics and Child Health, David Harvey Trust, Medical Women's Federation, Medact, Action Cerebral Palsy, Their World and Academy of Medical Sciences. NM is a member of the Nestle Scientific Advisory Board but accepts no personal remuneration for this role. NM has received research grants from the British Heart Foundation, Medical Research Council, National Institute of Health Research, Westminster Research Fund, Prolacta Life Sciences, Chiesi, Takeda, March of Dimes, Health Data Research UK, HCA International, and European Health Data Evidence Network; travel and accommodation expenses from Prolacta, Nestle and Chiesi. CG has received support from Chiesi Pharmaceuticals to attend an educational conference; in the past 5 years, he has been investigator on received research grants from Medical Research Council, National Institute of Health Research, Canadian Institute of Health Research, Department of Health in England, Mason Medical Research Foundation, Westminster Medical School Research Trust and Chiesi Pharmaceuticals. CB has received support from Chiesi Pharmaceuticals to attend an educational conference; she is supported by a personal award by the National Institute of Health Research.

**Patient and public involvement** Patients and/or the public were involved in the design, or conduct, or reporting, or dissemination plans of this research. Refer to the Methods section for further details.

**Patient consent for publication** Not applicable.

**Ethics approval** The Chief Investigator of the NNRD (NM) holds UK Research Ethics Committee approval (REC Reference: 16/LO/1093), and Confidentiality Advisory Group approval (ECC 8-05(f)/2010)) for the NNRD as a research database. Study-specific Research Ethics Committee (18/NI/0214), Health Research Authority and Health and Care Research Wales approval and approval for inclusion of their data in this study from all English neonatal units were obtained.

**Provenance and peer review** Not commissioned; externally peer reviewed.

**Data availability statement** Data are available upon reasonable request. Applications to use the data used within this project should be made to the Neonatal Data Analysis Unit, Imperial College London.

**ORCID iDs**
James Webbe http://orcid.org/0000-0001-8546-3212
Cheryl Battersby http://orcid.org/0000-0002-2898-553X
Chris Gale http://orcid.org/0000-0003-0707-876X

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
