## [Reviewer comments · BMJ Paediatrics Open]

This paper was submitted to a another journal from Archives of Disease in Childhood but declined for publication following peer review. The authors addressed the reviewers' comments and submitted the revised paper to BMJ Paediatrics Open. The paper was subsequently accepted for publication at BMJ Paediatrics Open.

ARTICLE DETAILS

TITLE (PROVISIONAL)	Use of parenteral nutrition in the first postnatal week in England and Wales: An observational study using real-world data
AUTHORS	Webbe, James Battersby, Cheryl Longford, Nicholas Oughham , Kayleigh uthaya, s Modi, Neena Gale, Chris

VERSION 1 – REVIEW

REVIEWER	Reviewer name: Dr. Shalini Ojha Institution and Country: University of Nottingham, Ireland Competing interests: None
REVIEW RETURNED	15-Mar-2022

GENERAL COMMENTS	Thank you for asking me to review this paper. This is a descriptive study of prevalence of parenteral nutrition (PN) use in England and Wales. The study is part of a larger work and the team have published other results previously. I have a few comments to make. This article describes practice during 2012-2017 and adds some descriptive data to our knowledge. However, the main results that PN use is higher in preterm infants and that there are geographic variations in use are well known. The information reflects practice between 2012-2017 but is unlikely to influence change or support decision making among clinicians. The article reads well, table are clearly labelled and laid out in a comprehensive, easy to follow manner. Statistical analyses are appropriate, and the authors have corrected for multiple comparisons in some tables. The variation in use across networks is expected but nevertheless interesting to see. Was there a difference in the population between these groups? Were there more preterm infants and/or larger numbers of sick term infants in some networks as compared to the others underlying the differences in PN use? Accounting for these could make the results more meaningful. Page 6 lines 8-12: Unfortunately, the BAPM And NICE recommendations do not match. Please specify which of these you have listed here.
---

	Page 11, lines 50-52: large proportion were term possibly because large proportion of all births are term and in any population that includes infants of all gestation, term infants outnumber preterms, especially very and extremely preterms by a large margin. This may be worth discussing especially as although large in numbers, term infants from only a very small proportion of those who receive PN.
--	--

VERSION 2 – Author’s Response

Manuscript ID bmjpo-2022-001543 - "Use of parenteral nutrition in the first postnatal week in England and Wales: An observational study using real-world data" Responses to reviewer comments

We are very grateful to the referee for their suggestions which we found useful. REVIEWER COMMENTS: Reviewer #1 Reviewer comment: This article describes practice during 2012-2017 and adds some descriptive data to our knowledge. However, the main results that PN use is higher in preterm infants and that there are geographic variations in use are well known. The information reflects practice between 2012-2017 but is unlikely to influence change or support decision making among clinicians. Response: The results of this study are similar to the results of previous work describing patterns of PN use internationally, but are the first time that this information has been made available to describe practice in the UK. Unfortunately delays in the writing and submission process mean that this data is now five years old, but we feel that it would still be of interest to clinicians as the most recent available data. The persistent geographic variation we have shown suggests optimal practice is unknown and that clinicians may practice relatively isolated from each other. Reviewer comment: The variation in use across networks is expected but nevertheless interesting to see. Was there a difference in the population between these groups? Were there more preterm infants and/or larger numbers of sick term infants in some networks as compared to the others underlying the differences in PN use? Accounting for these could make the results more meaningful. Response: These are important questions, but were beyond the scope of the work that was originally proposed. We do not have access to the denominator data required to undertake this analysis as we do not have a breakdown in different networks of births by gestational age or total numbers of unwell infants. Completing this work would require a new data extraction (needing new ethical approvals) and would be a new project. We hope to undertake this work (with more up to date data) in the future. We agree that this is a limitation of this work and have tried to make this more clear with the following edit (Page 13, Line 5): "From our descriptive analysis, it is not possible to determine whether the variation we identified is explained by regional differences in rates of prematurity, neonatal sickness or in other case-mix factors (such as congenital anomalies or neonatal surgical conditions), nor, given the paucity of evidence, is it possible to comment on what rate of PN use should be expected." Reviewer comment: Page 6 lines 8-12: Unfortunately, the BAPM And NICE recommendations do not match. Please specify which of these you have listed here. Response: To address this we have made it clear that the NICE guidance is most recent, and is the guidance to which we refer (Page 6; Line 1): "Following this, a national framework (12), and NICE guidance (13) have been developed. The most recent NICE guidance makes recommendations about prescription, administration, monitoring and recipients of PN in neonatal units; recommending all neonates born before 30+0 weeks+days gestation or weighing under 1250g at birth, and any who are unable (or not expected) to establish milk feeds of $\geq 100\text{mL/kg/day}$ by postnatal day five, receive early PN." Reviewer comment: Page 11, lines 50-52: large proportion were term possibly because large proportion of all births are term and in any population that includes infants of all gestation,

term infants outnumber preterms, especially very and extremely preterms by a large margin. This may be worth discussing especially as although large in numbers, term infants form only a very small proportion of those who receive PN. Response: We agree that the patterns identified are largely explained by the differences in births identified and have discussed this (Page 12, Line 50):
 “However, because of the much larger proportion of babies born at more mature gestations it is noteworthy that around 15% of all babies given PN are born at term

VERSION 3 – REVIEW

REVIEWER	Reviewer name: Dr. Peter Flom Institution and Country: Peter Flom Consulting, United States Competing interests: None
REVIEW RETURNED	14-Jun-2022

GENERAL COMMENTS	I mostly confine my remarks to statistical aspects of this paper. Unfortunately, I think the method is inappropriate and the conclusions mis-stated. A non-statistical point. I think the authors should define "parenteral nutrition". I didn't know the term so, of course, I Googled. Mostly, I got that it means "intravenous nutrition" but ... why not use "intravenous", which is surely a more common term? Is there some distinction here? p. 8 line 5 to 14 Categorizing continuous variables is OK for making tables and descriptions, but not good for modeling. It increases both type I and type II error and introduces a kind of "magical thinking" -- that something special happens at the cutpoints. I would leave GA and birth weight and year as numbers and use logistic regression. The DV would be "received PN" and the IVs would be GA, BW, year, and network. You could also use splines of the continuous variables to examine nonlinearities. This would also let you look at the relationships of each IV to the DV while controlling for the other IVs. line 17 Even if you decide to ignore my advice, above, using t-tests is not appropriate. There are multiple levels of both GA and BW. You can therefore use regression and have GA or BW as ordinal variables. But ordinal variables are tricky, this makes it even more sensible to not categorize. Table 1 makes the same point. How to interpret the fact that some categories are sig. and some not? Also, some have nonlinear trends. Here you could use logistic regression, with "received PN" as the DV and "year" and "GA" as the IVs. Year and GA could be treated as continuous measures, and a spline added. You might also want to look at the interaction. Table 2 - very similar comments as for table 1 Again, you could put tables similar to these in the final paper, but the p values should be based on logistic regression, and the results of the logistic regression should also be displayed. Finally, there's an issue with the conclusion that PN is "common". This ignores the fact that the population you are examining is not anything like the general population. Most babies are born at or close to term. Only about 10% of babies are pre-term, and most of those are close to full term. Only about 0.2% of term babies get PN, and only about 6% of near term babies get it.
---

	For instance, on p. 8 the authors state: 15% of neonates who received PN in the first postnatal week over the study period were born at term. This isn't wrong, but it is misleading. It is misleading in the same way as statement such as "90% of shark attacks occur in water that is 6 feet deep". Unless you state how much swimming occurs in such water (or, for your case, what proportion of babies are full term) this is very hard to interpret. The key fact is that very few (0.2%) of full term babies got PN. I am a statistician, not a neonatologist, but surely it's possible that 1 in 500 full term babies actually need PN. If you can make a case that NO full term babies should get PN, then you can do so. But, as is, I'm not sure the conclusions follow.
--	---

Version 3 – Author's Response

Manuscript ID bmjpo-2022-001543 - "Use of parenteral nutrition in the first postnatal week in England and Wales: An observational study using real-world data"

Responses to reviewer comments

We are very grateful to the referee for their suggestions which we found useful. We have reviewed the manuscript and following the comments feel that our attempt to describe how parenteral nutrition is used in clinical practice has been obscured by how we have displayed our data, thus we have undertaken some major revisions to make this important information clearer for neonatal clinicians. In particular we do not wish to model clinical practice for any purpose, but aim to make clinicians across the country aware of current practice.

REVIEWER COMMENTS:

Reviewer #1

Reviewer comment:

A non-statistical point. I think the authors should define "parenteral nutrition". I didn't know the term so, of course, I Googled. Mostly, I got that it means "intravenous nutrition" but ... why not use "intravenous", which is surely a more common term? Is there some distinction here?.

Response:

Parenteral nutrition is the more common terminology and is distinguishes it from any nutrition provided into the gastrointestinal system (enteral nutrition). It is a term that is common in neonatal medicine and would not need to be defined for our target audience.

Reviewer comment:

p. 8 line 5 to 14 Categorizing continuous variables is OK for making tables and descriptions, but not good for modeling. It increases both type I and type II error and introduces a kind of "magical thinking" --- that something special happens at the cutpoints. I would leave GA and birth weight and year as numbers and use logistic regression. The DV would be "received PN" and the IVs would be GA, BW, year, and network. You could also use splines of the contrinuous variables to examine nonlinearities. This would also let you look at the relationships of each IV to the DV while controlling for the other IVs.

Response:

We are grateful for this suggestion, but as discussed above our aim is not to undertake any kind of modelling. This modelling might tell of the relationship between different variables but is a different paper from the one we have written. Rather than trying to describe how an increase in birthweight/gestational age would affect the chances of a baby being given PN we want to allow current clinicians to see how their own practice compares to what is commonplace around the country: this

information is not currently available.

Furthermore, while we are categorising continuous variables, but we are using well established and accepted categories (using definitions from the World Health Organisation) that will be familiar to any practitioner within the field of neonatal medicine. We agree that these cut-points do not denote any special difference between categories, but they are how neonatologists are used to seeing results and will aid the interpretation of our results. To make this clearer we have made the following adjustment:

"For gestational age and birth weight, we grouped neonates according to well-established and widely accepted WHO categories (31, 32)."

Reviewer comment:

line 17 Even if you decide to ignore my advice, above, using t-tests is not appropriate. There are multiple levels of both GA and BW. You can therefore use regression and have GA or BW as ordinal variables. But ordinal variables are tricky, this makes it even more sensible to not categorize..

Response:

As discussed above our use of statistical testing was not optimal and has obscured our data. For this reason we have removed the T-test results, and furthermore removed all testing as we are aiming to describe current practice rather than test pre-specified hypotheses. Rather we present the data as it is, to allow our readers to draw their own conclusions. This can be seen in the extensive changes throughout the results section, and in the changes to Table 1-3.

Reviewer comment:

Table 1 makes the same point. How to interpret the fact that some categories are sig. and some not? Also, some have nonlinear trends. Here you could use logistic regression, with "received PN" as the DV and "year" and "GA" as the IVs. Year and GA could be treated as continuous measures, and a spline added. You might also want to look at the interaction.

Table 2 - very similar comments as for table 1

Again, you could put tables similar to these in the final paper, but the p values should be based on logistic regression, and the results of the logistic regression should also be displayed.

Response:

As above, we have removed the p values from the tables and removed accompanying text discussing them as we do not feel these results add to understanding of the raw data. We do not feel further modelling work would illustrate current practice any better.

Reviewer comment:

Finally, there's an issue with the conclusion that PN is "common". This ignores the fact that the population you are examining is not anything like the general population. Most babies are born at or close to term. Only about 10% of babies are pre-term, and most of those are close to full term. Only about 0.2% of term babies get PN, and only about 6% of near term babies get it.

Response:

We have clarified this comment to make it clear that parenteral nutrition use is common within the field of neonatal medicine

"PN is a common intervention on neonatal units and is given to 17% of all neonatal unit admissions."

Reviewer comment:

For instance, on p. 8 the authors state:

15% of neonates who received PN in the first postnatal week over the study period were born at term.

This isn't wrong, but it is misleading. It is misleading in the same way as statement such as "90% of

shark attacks occur in water that is 6 feet deep". Unless you state how much swimming occurs in such water (or, for your case, what proportion of babies are full term) this is very hard to interpret. The key fact is that very few (0.2%) of full term babies got PN. I am a statistician, not a neonatologist, but surely it's possible that 1 in 500 full term babies actually need PN. If you can make a case that NO full term babies should get PN, then you can do so. But, as is, I'm not sure the conclusions follow.

Response:

Neonatal doctors will be aware that term births are more common, and we are not attempting to mislead. We seek instead to highlight that while most clinicians associate PN use with small and preterm infants a large amount of PN gets given to term infants. As this is a population that are physiologically very different we wish to highlight that more thought should be given to how these babies should be cared for, and they should not simply be grouped with preterm populations. We have re-worded our discussion section as follows:

"However, because of the much larger proportion of babies born at more mature gestations it is noteworthy that around 15% of all babies given PN are born at term (although this still means that only 0.2% of term births receive PN). The energy requirements (36), indications for use and metabolic stability of these groups differs, thus the risks and benefits of PN may also differ substantially across gestational ages: guidelines and practice appropriate in one group may not be optimal in differing populations."